# Multi-Source Feature-Fusion Method for the Seismic Data of Cultural Relics Based on Deep Learning

**DOI:** 10.3390/s24144525

**Published:** 2024-07-12

**Authors:** Lin He, Quan Wei, Mengting Gong, Xiaofei Yang, Jianming Wei

**Affiliations:** 1Shanghai Advanced Research Institute, Chinese Academy of Sciences, Shanghai 201210, China; hel@sari.ac.cn; 2University of Chinese Academy of Sciences, Beijing 100049, China; 3Sichuan Museum, Chengdu 610000, China; 13501868706@163.com (Q.W.); 09210540008@fudan.edu.cn (M.G.)

**Keywords:** cultural relics conservation, cultural relics seismic damage, event ontology, multi-source information fusion, deep learning

## Abstract

The museum system is exposed to a high risk of seismic hazards. However, it is difficult to carry out seismic hazard prevention to protect cultural relics in collections due to the lack of real data and diverse types of seismic hazards. To address this problem, we developed a deep-learning-based multi-source feature-fusion method to assess the data on seismic damage caused by collected cultural relics. Firstly, a multi-source data-processing strategy was developed according to the needs of seismic impact analysis of the cultural relics in the collection, and a seismic event-ontology model of cultural relics was constructed. Additionally, a seismic damage data-classification acquisition method and empirical calculation model were designed. Secondly, we proposed a deep learning-based multi-source feature-fusion matching method for cultural relics. By constructing a damage state assessment model of cultural relics using superpixel map convolutional fusion and an automatic data-matching model, the quality and processing efficiency of seismic damage data of the cultural relics in the collection were improved. Finally, we formed a dataset oriented to the seismic damage risk analysis of the cultural relics in the collection. The experimental results show that the accuracy of this method reaches 93.6%, and the accuracy of cultural relics label matching is as high as 82.6% compared with many kinds of earthquake damage state assessment models. This method can provide more accurate and efficient data support, along with a scientific basis for subsequent research on the impact analysis of seismic damage to cultural relics in collections.

## 1. Introduction

With the frequent occurrence of seismic disasters, there is an urgent need for seismic preventive protection of cultural relics in collections. As an important part of human history and cultural heritage, the collection of cultural relics will suffer immeasurable losses if damaged by seismic activity, as the relics are both fragile and vulnerable [1,2]. At present, there is still uncertainty about the impact of seismic damage on cultural relics. Damage impact factor analysis and research into the effects of seismic damage on collections of cultural relics, as the theoretical support of cultural relics seismic disaster risk assessment and prediction, is the premise of effective cultural relics seismic preventive protection work. However, the seismic damage influencing factors of the cultural relics in the collection is multi-sourced, and it is difficult to obtain information about the cause; moreover, the real data of the cultural relics in the collection at home and abroad are very limited, and the data sources are complex and varied in type [3,4]. The lack of relevant real data, along with the lack of effective processing methods, seriously restricts the progress of the research on the impact of seismic damage on the cultural relics in the collection. Therefore, there is an urgent need to study the acquisition and processing methods of seismic data for cultural relics in collections and to form a dataset that can be used to analyze the impact of seismic damage on cultural relics in collections. This is carried out for the following reasons: to explore the damage mechanism of the seismic impact, formulate efficient preventive protection measures for collection relics, provide a scientific basis and reference for the seismic protection of the museum system, and contribute to the long-term preservation of cultural relics and cultural heritage.

The seismic data of the cultural relics in the collection not only include the ontological properties of the cultural relics in the collection but also involve their preservation space, exhibition facilities, and seismic information at the same time and space. Due to the uncertainty and huge damage caused by seismic activity, the acquisition of data on the seismic damage caused to cultural relics in collections is complicated. First of all, due to the scarcity and high value of the collected cultural relics, the acquisition of their physical property information is difficult to directly measure by contact experiments [5]. Only the appearance and structure information of the cultural relics’ bodies, including their basic attribute data such as size, weight, and material, can be collected by basic measuring tools such as rulers, balances, or chemical analysis [6]. The high-precision geometric shape, texture, and color information of cultural relics are usually acquired by non-contact methods such as laser scanning or high-definition filming [7,8]. This process involves the application of data from multiple sensors. However, the preservation space and exhibition facilities of cultural relics are related to the preservation and exhibition conditions of museums, which usually need to be acquired by fieldwork statistics [9]. Additionally, some of the composite attributes, such as the center of mass of cultural relics and the power amplification factor of exhibition cabinets, are usually calculated by empirical formulas or experimental analysis [10]. Secondly, the seismic-related data, including the magnitude, intensity, duration, and other information, need to be obtained by researching the local historical records of seismic occurrence [11]. Finally, the construction of the dataset needs to consider the definition of the categories and attributes of the data items; however, currently, there is a lack of category delineation systems related to cultural relics datasets, which makes it difficult to form a logical and hierarchical complete dataset [12,13]. At the same time, for the collection of cultural relics that have been collected, seismic data still need to be labeled and categorized. However, with the continuous development of computer technology, methods based on deep learning [14] and mathematical statistics [15,16] have been applied to the task of multi-source data processing. The measurement of relic data requires not only safety but also a certain level of expertise. However, there is a lack of relevant researchers at present, and it will waste a lot of labor costs. Meanwhile, the work of manual subjective statistics greatly affects the efficiency and quality of seismic data processing of cultural relics in the collection [17]. In summary, the collection and processing of data concerning seismic damage to cultural relics in collections still raises many problems that have not been solved.

To address the abovementioned problems concerning seismic data collection and the calculation of cultural relics in the collection, we propose a number of effective solutions.


**Contributions:**


Formulate a multi-source data-processing strategy for seismic damage of cultural relics in collections. By defining the seismic damage event-ontology model; sorting out the relationship between the event attributes of the cultural relics in the collection; and designing an empirical model to solve the dynamic coefficient, center of mass, and other event attributes of cultural relics exhibition facilities, a universal and standardized seismic damage data index system for curatorial cultural relics is proposed.Propose a multi-source feature-fusion matching method based on deep learning, as well as the fusion of superpixel map convolution, to assess the damage status of seismic-damaged cultural relics. This is combined with deep learning fusion of multi-source information to realize automated annotation of massive cultural relics seismic damage image data to improve data quality and processing efficiency.Construct a complete dataset for the analysis of the impact of seismic damage on cultural relics in the collection. Based on a variety of seismic damage cultural relics data acquisition and automation methods, we processed the seismic damage data of 1352 cultural relics in the collection and formed an accurate, comprehensive, and standardized seismic damage dataset of cultural relics in the collection.

## 2. Related Works

### 2.1. Data Processing Strategy—Related Works for Cultural Relics

The measurement and analysis of the properties of cultural relics is the basis of cultural relics protection work. With the continuous development of science and technology, the combination of artificial intelligence and Internet of Things technology for data collection and processing has gradually become an important research direction in the field of cultural relics protection [18,19]. Many new technologies and new methods are gradually being applied to the statistics and analysis of cultural relics information and achieve more significant results [20,21].

In order to avoid causing damage to cultural relics during the measurement process, researchers have carried out non-destructive or minimally destructive measurements of cultural relics by introducing advanced measurement techniques such as hyperspectral imaging, laser scanning, and X-ray fluorescence (XRF) [22]. Xi’an Museum [23] provides panoramic spectral image data for cultural relics conservation and research through multi-camera XRF image stitching. Masi [24] and Anger et al. [25] used XRF to treat the detection of cultural relics for analysis and to determine the age of their appearance and the materials used. Zhang et al. [26] examined the traditional method of single-point monitoring and a low degree of automation and other issues, as well as the use of three-dimensional laser scanning technology for long-term monitoring of natural cultural relics erosion and spalling. Xu [27] et al., through micro-area X-ray analysis of bronze cultural relics composition and process, targeted the selection of restoration methods and materials. In order to obtain the basic properties of cultural relics more accurately and quickly, the Palace Museum constructed the “conceptual reference model of ancient Chinese movable cultural relics” [28]. The statistical method of seismic damage information of cultural relics is often through manual research and recording. Although this method is cumbersome, its statistical accuracy is high [29]. In addition, theoretical support is also provided for the processing of cultural relics’ seismic damage information through methods such as artificial intelligence and experimental simulation [30]. Yu et al. [31] studied and explored the implicit information mining method based on an autoencoder to realize the extraction of risk relationships among movable cultural relics. Qin et al. [32] used the proposed static method to derive the stability calculation formula of three destabilization modes and analyzed the influence of seismic influence coefficients, such as the crack bond strength, on the stability of cultural relics. In order to study the dynamic characteristics as well as the seismic performance of wooden buildings, Zhao et al. [33] analyzed the vibration frequency of internal cultural relics with the Yuhua Pavilion of the Forbidden City as an analytical object. Ji et al. [34] developed a linear guide-type horizontal seismic isolation device for cultural relic display cabinets according to the museum display cabinets and tested the self-vibration characteristics of the seismic isolation system for cultural relic display cabinets using horizontal unidirectional seismic waves.

Although there are various ways to process cultural relics data, their standards are inconsistent and lack standardization. Most of the current methods for processing seismic damage information on cultural relics are aimed at immovable cultural relics. Most of the processing of seismic damage information of cultural relics in collections focuses on the conceptual design based on the expert’s empirical knowledge, the vibration test analysis guided by the theory of structural dynamics, and numerical simulation research based on simulation modeling. However, these methods are mostly carried out in experimental environments, based on certain assumptions and simplification principles, and cannot completely reproduce the seismic scene and the damage process of cultural relics. At the same time, because of the non-renewable nature of cultural relics and their precious cultural value, the research is usually based on alternative samples as experimental objects, which are different from real cultural relic materials. This is bound to cause bias in the experimental results, which is also an important reason for the current difficulty in popularizing personalized seismic safety measures for cultural relics. Therefore, it is particularly important to study a method of processing the real seismic data of cultural relics in a collection.

### 2.2. Multiple-Source Data Fusion—Related Works

The seismic damage data of cultural relics in collections is important for the in-depth study of the damage mechanism of cultural relics in seismic, the assessment of the risk of seismic damage to cultural relics, as well as the development of effective anti-seismic protection measures [35]. The seismic damage data of cultural relics contain multi-source information such as images, text, tables, etc., which means that it is difficult to fully explore the potential characteristics of the data using a single type of data-processing method [36]. Therefore, it is necessary to carry out relevant research in combination with multi-source data feature fusion and extraction.

In order to fully utilize the features of multi-source data and achieve their deep fusion, Tong [37] and Miao et al. [38] proposed a coding method for multi-modal data, which achieves multi-modal feature fusion through multi-layer neural network classifiers and probabilistic maps. Yan [39] and Chen et al. [40] proposed corresponding structured and unstructured data extraction and data-storage methods for multi-source heterogeneous data, respectively, to realize feature-level data fusion of multi-source heterogeneous big data. Xu [41] and He et al. [42] integrated the heterogeneous features of multi-source data and used a weighted adaptive-fusion method to realize feature fusion. In the application of multi-source data in distribution networks, Huang [43] and Wu et al. [44] proposed a correlation method based on multi-source heterogeneous data to use monitoring data as network inputs to realize the operational state assessment of distribution systems. He [45] and Jin et al. [46] integrated multi-source drug information through cross-view comparative learning and multi-source feature fusion, respectively, and drug interaction prediction was realized by encoding and fusing features. Deep learning-based network models are effective in multi-source data feature fusion. Zhang [47] and Zhang et al. [48] proposed a multimodal feature-fusion method for image and text data, using convolutional neural networks and long- and short-term memory stretching networks to achieve embedding of image and text data. Similarly, Islam et al. [49] used a convolutional block attention module and a convolutional long short-term memory network to process image and time-dependent multi-source sensor information. Tian [50] used batch-normalized long short-term memory and a three-bit convolutional transformer based on a self-attention mechanism for feature extraction and fusion based on multi-source heterogeneous data in industrial surveillance. Liu et al. [51] proposed a regional flow pattern knowledge mining network by combining multi-source heterogeneous big data with feature modeling in order to mine and store urban flow patterns. To improve the prediction accuracy of existing models, Strelet [52] and Liu et al. [53] classified and cleaned the multi-source data, respectively. Wang et al. [54] used the long sequence formed by splicing the samples of multi-source heterogeneous data as the input of the prediction model, which effectively improves the prediction accuracy of the risk-assessment model.

Although there are rich methods oriented to the processing of heterogeneous data from multiple sources, the inconsistency between the data makes it easy to produce misleading results after feature fusion. Missing values and outliers contained in the data also affect the accuracy of the analysis results. At the same time, due to the seismic and damage to the collection of cultural relics having more factors, the existing feature-extraction and -fusion methods have poor scalability. The future of large-scale, complex cultural relic data-processing analysis work faces issues such as being time-consuming, laborious, having low accuracy, and other issues. Therefore, it is necessary to strengthen the research on the multi-source heterogeneous data of cultural relics to improve the quality of data processing and analysis efficiency.

## 3. Multi-Source Data-Processing Strategy for Seismic Damage of Cultural Relics

We take China’s specification for seismic protection of cultural relics in collections (WW/T0069-2015) [55] as the benchmark index and combine it with the latest integrated whole-system seismic protection design concepts to establish an event-ontology model for the analysis of seismic impacts on cultural relics in collections. In this section, we clarify the elements and logical relationships of seismic impact events of cultural relics in the collection and design the data-attribute processing strategy for the difficulties in acquiring data on cultural relics ontology and exhibition and collection facilities.

### 3.1. Cultural Relic Seismic Impact Analysis in the Event-Ontology Model

Due to the multiple sources and complex correlations involved in cultural relics seismic data, it is difficult to utilize them directly for cultural relic management and seismic disaster protection. Traditional databases can only play the role of storage and retrieval, and although they can search and organize cultural relics data more accurately and conveniently, they can not quickly discover and reason about the correlation between various attributes of cultural relics. Therefore, in order to improve the standardized management level of the database and to carry out cultural relics disaster protection in a more comprehensive, complete, and systematic way, it is necessary to construct a scientific and reasonable cultural relics data storage architecture. Based on this, we construct the Event Ontology Model for Seismic Damage Impact Analyzing on Cultural Relics Collections (EOM-SR).

As shown in Table 1, the attributes of the seismic damage events of cultural relics in collections are defined. In the Event Ontology Model for Seismic Damage Impact Analysis on Cultural Relics Collections, the seismic damage impact on cultural relics in the collection is defined as an event. The event defines the attributes of the seismic impact event of the curatorial cultural relics, which mainly includes event elements, event relationships, and event types. The event elements are divided into subject, object, influence factors, time, and space. The influencing factors of the event are divided into the ontological state of cultural relics, exhibition and collection facilities information, preservation space information, and seismic information. The event relationship is divided into a cause-and-effect relationship, containing relationship, time relationship, and space relationship.

Based on the definition of seismic event attributes, we construct EOM-SR based on the simple event model, as shown in Figure 1. The EOM-SR contains all the attributes of seismic events. Among them, the circular entities represent the main attributes such as subject and object, and the connecting lines indicate the inclusive relationship between the entities’ attributes.

The EOM-SR creation process is shown in Figure 2. First, the seismic damage information and cultural relics attributes involved in the seismic damage events of the cultural relics in the collection are counted. After that, the seismic events according to the seismic event attributes of the cultural relics in the collection are judged, and the ontology attributes of the seismic events are extracted if they meet the defined contents. Finally, the event-ontology model is constructed to be used for subsequent correlation analysis and evaluation of seismic damage data of cultural relics in the collection.

The collection process of seismic damage information of the cultural relics in the collection requires the staff to have rich professional knowledge and practical experience. Usually, after the seismic event, the cultural relics department organizes experts and technicians to go to the disaster area to conduct an on-site investigation by assessing the degree of damage to cultural relics, damage causes, and other preliminary information collection and record the detailed associated survey information. The seismic damage data of the cultural relics in the collection involves many detailed attributes, and the content is relatively rich. Based on the empirical knowledge, we mainly count the data attributes involved in the elements of seismic damage of cultural relics in the collection and categorize them according to the four first-level categories of Class (cultural relics ontology, cultural relics exhibition facilities, cultural relics preservation space, and seismic information). Under each classification, there is a second-level classification $Class_{i}$, i.e., cultural relics seismic damage data attributes (see Appendix A, Table A1).

Although there are large differences in the relevant data items of different custodian units, there are only 56 units of damaged cultural relics in this seismic (the 56 units are all from Sichuan Province, China, and include different sizes of museums and cultural relics preservation units). Its data volume is relatively small, and the corresponding storage space attributes of cultural relics in the same storage space are the same. Therefore, in this paper, the data items under the category of cultural relics preservation space are mainly counted using a questionnaire (see Appendix A, Table A2 and Table A3). In addition, the relevant data items are organized into a table and uniformly summarized into the seismic damage database of cultural relics in collection.

Seismic information is the seismic information of the site where the damaged cultural relics custodian unit is located. This information is mainly captured from publicly available seismic historical data. It should be pointed out that the seismic information related to damaged cultural relics currently collected in this dataset is under the same seismic effect and has the same time characteristics. The damaged cultural relics are located in the scattered distribution of cultural and museum units with a large spatial span, which has a clear spatial–temporal characteristic. In this study, nine kinds of data items are selected as the collection data items of seismic information.

We combine the seismic information of the global strong seismic spatial and temporal distribution dataset from 1989 to 2018 with the statistical seismic data based on the geographic locations of existing cultural heritage collection units. We record the geological conditions of 56 cultural relics units based on the addresses where their buildings are located. Moreover, combined with the strong global seismic spatial–temporal distribution data, we query the seismic events that have an impact on the cultural relics and record the corresponding seismic moment, magnitude, and other data items of each cultural relics collection unit. We calculate the seismic source coordinates for the 5.12 Wenchuan seismic.

As shown in Figure 3 for the global strong seismic spatial and temporal distribution dataset visualization display effect, each point on the Earth represents a seismic source. The gray box in Figure 3 shows the information related to the Wenchuan seismic, and the blue box on the right side shows the retrieved information related to the seismic in the Sichuan Museum. Detailed information such as latitude, longitude, and seismic intensity of the Wenchuan seismic source can be directly queried using AriGIS software (version 10.2). By virtue of the software, we can query the coordinates of the cultural relics storage unit and the coordinates of the seismic source, the moment of the seismic event, and other information for the subsequent calculation of the seismic damage to cultural relics information.

After that, the distance from the epicenter is calculated based on the seismic source information. Since the Earth is a celestial body approximating a sphere, the distance information is estimated by combining the coordinates of the calculated curatorial units with the latitude and longitude approximation of the seismic event. The distance from the epicenter between two points is an arc on the Earth’s surface and is calculated as shown in Equations (1)–(3).
(1)$$a=\sin^{2}\left(\frac{\Delta \mathrm{lat}}{2}\right)+\cos\left({\mathrm{lat}}_{1}\right)\cdot \cos\left({\mathrm{lat}}_{2}\right)\cdot \sin^{2}\left(\frac{\Delta \mathrm{lon}}{2}\right)$$
(2)$$c=2\cdot \mathrm{atan}2(\sqrt{a},\sqrt{1-a})$$
(3)distance =R⋅c
where Δlat is the difference between the latitude of the second point and the latitude of the first point; Δlon is the difference between the longitude of the second point and the longitude of the first point; ${\mathrm{lat}}_{1}$ and ${\mathrm{lat}}_{2}$ are the latitudes of the first and second points, respectively, and R is the radius of the Earth.

### 3.2. Ontology of Cultural Relics and Exhibition Facilities Data Attribute Processing

When assessing the mechanisms of earthquake damage to artifacts in a collection, a variety of measurement sensors were used to obtain data on the artifacts (for example, in the measurement of relics, we used equipment that meets international standards, such as scales and electronic scales. We used position sensors to measure the position information of the artifacts on display. Image sensors were used to capture image information. Infrared sensors were used to determine the damage status of the artifacts). These data serve as important input features for the study. Due to different cultural relics, ontology attributes and exhibition facilities are different, and the number of cultural relics in the collection is large and difficult to count one by one. Therefore, the need for the above data items in the existing text, tables, and other cultural relics attribute data in accordance with the unified specification for normalization. Data cleaning was used to find and deal with missing values, abnormal values, or error values to ensure data quality. After that, the seismic damage related data items that cannot be directly obtained were calculated according to the empirical formula to improve the seismic damage data information of cultural relics. The specific process is shown below.

Step 1: Data collection of cultural relics ontology and exhibition and collection facilities. Due to the different information-collection requirements of cultural relics storage units, there are some missing data items to be counted. Specific treatment for the directly accessible information on cultural relics includes the size of cultural relics, quality and other numerical categories of missing items, the use of manual methods of measurement, and statistics. Figure 4 shows the size and quality data-measurement process.

For the cultural relics, storage location, exhibition and storage facilities, and other text-based data were determined through the template-matching method. The existing database/table query and storage, as shown in Figure 5, was used for the cultural relics exhibition and storage facilities information-query process. The blue part on the left represents the logical process of cultural relic information retrieval, and the right side is the actual data-transfer process of the template-matching algorithm. It extracts the keywords for heritage retrieval based on template matching triggering wildcards (“*” stands for a wildcard. The areas delineated by the dotted lines represent the different steps, while within the same area are the methods and implementation techniques corresponding to the steps). After that, it realizes heritage information retrieval and matching using template query based on existing heritage databases and tables.

Step 2: Data item preprocessing. Since there may be differences in the data formats and standards obtained by different equipment and different collection methods, it is necessary to eliminate these differences through data standardization. Through this operation, all the data follow unified norms and standards to ensure the accuracy and comparability of the data, laying the foundation for the subsequent analysis of cultural relic seismic data. Specific treatment for the existing data unit metrics is performed in accordance with the table of data attributes for uniform conversion—for example, for the uniform conversion of units of length to centimeters (cm), the weight of artifacts is measured in grams (g). Area and volume are recorded using square centimeters (cm^2^) and cubic centimeters (cm^3^), respectively.

Among them, the missing values rely on the information template-matching query shown in Figure 5 to directly locate the cultural relics data items that have not been captured or recorded so as to facilitate the re-verification of cultural relics information. Since the range of differences in size and dimensions of the artifacts in the collection is small, the box-plot method is used for the detection of outliers. Firstly, we can retrieve the data items of the same class of cultural relics and obtain the set $N(n_{1},n_{2},n_{3},\cdots ,n_{k})$ after sorting according to the data size, and then obtain the values of the third quartile Q3 and the first quartile Q1, which are calculated as shown in Equations (4) and (5):(4)$$Q1=\left\{\begin{array}{c} \begin{array}{cc} n_{\left(\frac{n+1}{4}\right)}, & \frac{n+1}{4}\in N \end{array}\\ \begin{array}{cc} \frac{n_{\left(\frac{n-1}{4}\right)}+n_{\left(\frac{n+1}{4}\right)}}{2}, & \frac{n+1}{4}\notin N \end{array} \end{array}\right.$$
(5)$$Q3=\left\{\begin{array}{c} \begin{array}{cc} n_{\left(\frac{3n+3}{4}\right)}, & \frac{3n+3}{4}\in N \end{array}\\ \begin{array}{cc} \frac{n_{\left(\frac{3n-3}{4}\right)}+n_{\left(\frac{3n+3}{4}\right)}}{2}, & \frac{3n+3}{4}\notin N \end{array} \end{array}\right.$$

We can calculate the quartile distance IQR and the upper and lower box limits Ub and Lb from Q1 and Q3. This is shown in Equations (6)–(8).
(6)IQR=Q3−Q1
(7)Ub=Q3+1.5×IQR
(8)Lb=Q3−1.5×IQR

Possible outliers are identified and recaptured by looking for data points in the dataset that fall outside the range. Finally, the range is harmonized according to the unit of measure.

Step 3: Composite attribute data item calculation. The stability of cultural relics in the collection is one of the important factors in the study of the impact of cultural relics on seismic damage, while the center of mass of cultural relics and the power amplification factor are closely related to the stability of cultural relics. Since these two data items can not be obtained directly, the empirical formula method is used for calculation. The center of mass coordinates $\left(x_{c},{\;y}_{c}{,\;z}_{c}\right)$ of the cultural relics are calculated as shown in Equations (9)–(11).
(9)$$x_{c}=\sum_{i}\left(m_{i}\cdot x_{i}\right)/\sum_{i}m_{i}$$
(10)$$y_{c}=\sum_{i}\left(m_{i}\cdot y_{i}\right)/\sum_{i}m_{i}$$
(11)$$z_{c}=\sum_{i}\left(m_{i}\cdot z_{i}\right)/\sum_{i}m_{i}$$
where $\left(x_{i},y_{i},z_{i}\right)$ is the coordinate of the i-th mass point on the artifact. The method is based on a discrete approximation of the mass distribution of the artifacts. Since the overall material of the artifacts tends to be the same in the actual center of mass calculation process, it can be regarded as a continuous distribution of the mass, and the integral can be used to calculate the location of the center of mass.

The dynamic amplification factor (Dynamic Amplification Factor, DAF) of the cultural relics cabinet frame depends on the intrinsic frequency of the structure, damping ratio, seismic input frequency, and other factors. In order to facilitate the calculation, we combined the properties of the existing cultural relics, assuming the cultural relics cabinet rack is a single-degree-of-freedom system in which the mass of the mass point is m, the stiffness is k, the damping ratio is e, the frequency of the seismic input is $\omega_{\mathrm{input}\;}$, and $\omega_{0}$ is the intrinsic frequency of the structure. Equations (12)–(14) show the calculation process of the dynamic amplification factor DAF.
(12)$$DAF=\frac{1}{\sqrt{{\left(1-\beta^{2}\right)}^{2}+{\left(2\xi \beta \right)}^{2}}}$$
(13)$$\beta =\frac{\omega_{\mathrm{input}\;}}{\omega_{0}}$$
(14)$$\omega_{0}=\sqrt{k/m}$$

Through the processing and attribute calculation of cultural relics seismic damage data using the above methods, the collected multi-source heterogeneous curatorial data are attribute-extracted according to the ontology model of cultural relics seismic damage events. They are normalized and stored according to the event elements to form a universal and standardized indicator system for seismic damage data of cultural relics in the collection.

## 4. Multi-Source Feature-Fusion Matching of Cultural Relics Based on Deep Learning

Aiming at the problem wherein the damage status of cultural relics in the collection is currently labeled manually and subjectively and lacks objective assessment, we proposed a method of evaluating the damage status of cultural relics by fusing the convolution of superpixel maps so as to improve the accuracy of the data using the damage status of cultural relics. At the same time, for the efficient and accurate association and matching of multi-source heterogeneous data of massive cultural relics, we propose an automatic matching method between cultural relic images and text data to improve the efficiency of multi-source data fusion of seismic impacts of cultural relics in the collection.

### 4.1. Damage State Assessment Method for Cultural Relics Based on Superpixel Map Convolution

At present, most of the post-seismic cultural relic damage-determination work adopts the manual method, which requires a certain professional knowledge reserve to realize the damage level classification of the seismic-damaged cultural relics. Moreover, there is no standard definition for the conclusion of damage determination, and there exists a certain subjective component. Since there is no standard for the damage level of cultural relics, we classified it into five damage statuses according to the seismic damage of cultural relics (as shown in Table A3 in Appendix A, including radioactive damage, partial damage, fracture, cracking, deformation, scratches, and scratches).

In order to improve the standardization, accuracy, and practical efficiency of the statistical work of seismic damage information of cultural relics and to improve the seismic damage dataset of cultural relics in collections, we propose a Damage State Assessment Method for Cultural Relics based on Superpixel Map Convolution (ASMC) to achieve efficient and objective assessment of the damage level of cultural relics. The architecture of the assessment method is shown in Figure 6. It mainly adopts a two-branch structure to recognize the damage degree of cultural relics by inputting the original image and the post-damage image into the model. The post-damaged artifact image is divided into multiple regions via superpixel map segmentation of the artifact’s ontology. After that, high-level semantic information is extracted by deep map convolution. Finally, the deep feature map fusion with the original image is used to guide the classification of artifacts into shock-damaged categories. The detailed realization steps are as follows.

First, the deep residual block network branch is constructed to extract the deep features of cultural relics. Aiming at the network degradation problem of the deep model, the residual block structure is designed to ensure the transmission of artifact features. As shown in Equations (15) and (16), the input artifact image Img is operated after convolution and pooling for feature $Map_{0}$, and then the deep semantic feature extraction is carried out by four residual blocks $Block_{i}$. The corresponding block feature map is output after each residual block calculation $Map_{i}$. During the computation of each residual block, we use a convolutional kernel of 3 × 3 size for feature dimensionality reduction to extract semantic features at different levels.
(15)$$Map_{0}=Pooling(Conv_{3\times 3}(Img))$$
(16)$$Map_{i}=\mathrm{Relu}(\mathrm{Relu}(Conv_{3\times 3}(\mathrm{Relu}(Conv_{3\times 3}(Map_{i-1}))))+Map_{i-1})$$

As shown in Equation (17), the output feature map $Map_{4}$ is dimensionally unified by the convolution operation to obtain the artifact feature vector $V_{r}$ used for guiding the artifact state class classification.
(17)$$V_{r}=Conv_{1\times 1}(Map4)$$

Second, a network branching based on a hyperpixel map is proposed to extract features of seismic-damaged cultural relics. Due to the differences in different parts of the artifact ontology after the seismic damage, it is difficult to correctly assess the damage of different regions of the artifacts by directly using the convolution structure for feature extraction. Therefore, as shown in Equations (18)–(20), we adopt the Slic image segmentation method to divide the image into regions and construct a graph convolutional neural network to mine the regional features based on the artifact hyperpixel map $Img_{seg}$. During the superpixel map computation, the output feature map size is the square of the number of channels corresponding to the residual feature map $Map_{i}$. The output feature vector passes through the SoftMax after feature fusion to assess the grade of the artifact shock damage status.
(18)$$Img_{seg}=Slic_{16}(Img_{s})$$
(19)$$V_{i}=Graph_{conv}(Conv_{3\times 3}(Map_{i-1}\times V_{i-1}))$$
(20)$$V_{s}=Conv_{1\times 1}(V_{4}\times Map_{4})$$

### 4.2. Automatic Matching Method Based on Multi-Source Information Fusion

Due to the diverse sources of seismic data of cultural relics, how to effectively integrate these data and ensure the accuracy and reliability of data matching (especially the matching of cultural relics’ images and structured data) is an important issue facing the processing of seismic data of cultural relics. At present, there is no data-integration method applicable to cultural relics data. With the continuous accumulation of data, methods such as manual processing require a large amount of time costs and are inefficient and difficult to use to meet the needs of real-time analysis and rapid response. To solve this problem, we propose an Automatic Matching Method based on Multi-source Information Fusion (A3MIF) for cultural relic seismic data. The method consists of three parts: semantic feature representation, visual feature learning, and multi-source information fusion, which is able to enhance the relevance of labels and image regions through a semantically guided attention mechanism by simultaneously utilizing semantic labels and visual information of cultural relics. Meanwhile, the semantic information is used to generate a semantic dictionary with tag relevance constraints to reconstruct the visual features. Finally, the normalized representation coefficients are obtained as the probability of the artifact image classification for data matching. The overall framework of A3MIF is shown in Figure 7.

First, the semantic dictionary of cultural relics seismic damage with relevance constraints is learned through the semantic feature representation module to realize the deep semantic embedding of cultural relics description statements. In this study, the semantic information of cultural relics labels is used to generate semantic dictionaries aligned with the visual space, and a multilayer GRU neural network is used to obtain semantic embedding vectors for each cultural relic category label $d_{s}(d^{0},d^{1},d^{2},\cdots ,d^{i})$. For a given text sequence $x_{\left(i\right)}$, hidden state $d^{\left(i-1\right)}$, and gating function, the update formulas of the GRU are shown in Equations (21)–(23).
(21)$$z^{\left(i\right)}=\sigma \left(W_{z}\cdot \left[d^{\left(i-1\right)},x^{\left(i\right)}\right]+b_{z}\right)$$
(22)$$r^{\left(i\right)}=\sigma \left(W_{r}\cdot \left[d^{\left(i-1\right)},x^{\left(i\right)}\right]+b_{r}\right)$$
(23)$${\tilde{d}}^{\left(i\right)}=\tanh\left(W_{h}\cdot \left[r^{\left(i\right)}⊙d^{\left(i-1\right)},x^{\left(i\right)}\right]+b_{h}\right)$$
where the output of the update gate is $z^{\left(i\right)}$, the candidate hidden state is ${\tilde{d}}^{\left(i\right)}$, σ is the Sigmoid function, tanh is the hyperbolic tangent function, W is the weight matrix, and b is the bias vector. The long-term dependencies in the artifact dictionary sequence can be effectively learned and preserved by multi-layer GRU and avoid the gradient vanishing problem in traditional RNN.

After that, a deep convolutional neural network (CNN) is used in the visual feature learning part to extract the key features of the cultural relics seismic damage images. In this study, a deep feature extraction network architecture based on CNN is designed for the input images of cultural relics with seismic damage, and the output feature maps containing high-level semantic information are obtained after three convolutions and two down sampling iterations. In the feature-extraction process, the size of convolution kernel size is 3 × 3. The computational process is shown in Equation (24).
(24)$$M_{W\times N\times H}=Pool(Conv(Input_{3\times 512\times 512}))$$

Finally, in the multi-source information fusion part, semantic tags are embedded into the artifacts’ seismic damage feature maps using a semantically guided attention-based mechanism. The semantic feature compression is performed on the artifact feature map using the channel attention mechanism, and the embedding representation $d_{i}$ based on the semantic dictionary of artifacts is dimensionally aligned using the fully connected network Flatten to obtain the feature vector containing the damaged image of artifacts $F_{Map}$. The multisource information fusion with semantic feature representation and visual feature map is achieved by adding the artifact dictionary embedding vector $d_{i}$ correspondingly with $F_{Map}$. The embedding vector $d_{i}$ is used as the embedding vector for the artifacts and the visual feature map. The predicted probabilities are calculated by the Softmax layer for label matching in the cultural relics seismic damage database. The realization process is shown in Equations (25) and (26).
(25)$$F_{Map}=Flatten\left(Conv_{1*3}\left(GAP\left(M_{W\times N\times H}\right)\right)\right)$$
(26)$$F_{fu}=\mathrm{SoftMax}(F_{Map}+d_{i})$$

### 4.3. Loss Function

The process of relics seismic damage class classification and data matching based on deep learning mainly involves the training problem of the classification model. Since the classification of the cultural relics seismic damage class is used to classify multiple degrees, we adopt the cross-entropy loss function $L_{c1}$, which can measure the difference between the predicted probability distribution and the true probability distribution well. The calculation process is shown in Equation (27).
(27)$$L_{c1}=-\sum \left(y_{i}\cdot \log(p_{i})\right)$$
where $y_{i}$ denotes the value of the *i*-th class in the real label, which usually takes the value of 0 or 1 for multi-categorization problems, and only one class is 1 (which means that the class is the real class), and the rest of the classes are 0. $p_{i}$ denotes the probability that the model predicts the *i*-th class (i.e., the output after the Softmax function). For relic data matching, the model is trained with the binary cross entropy loss function $L_{c2}$ as shown in Equation (28).
(28)$$L_{c2}=-[y\log y_{i}+(1-y)\log(1-y_{i})]$$

Among them, y denotes the true label, and $y_{i}$ denotes the probability predicted by the model. The relic classification and matching models are trained by two loss functions, and the related results are analyzed in Section 5 and Section 6.

## 5. Experimental Analysis

### 5.1. Experimental Environment and Dataset

In terms of the experimental environment and parameter settings for automatic matching of artifact seismic data, the input image data were randomly clipped to a (3, 512, 512) size input. The word-embedding vector dimension size of the artifact dictionary is 300, and the Adam optimizer is used for parameter training and gradient updating. The model uses L2 regularization to prevent overfitting. The experimental environment is implemented using the deep learning framework PyTorch, and the relevant information is shown in Table 2.

This study uses multi-source information fusion to realize the image matching of cultural relics, but the public dataset of cultural relics is smaller. Therefore, in order to evaluate the effectiveness of A3MIF, we built our own Seismic Damage Dataset of Cultural Relics (CR-SDD) [56] based on the provided cultural relics data for training and testing the model. At the same time, two datasets, VOC2012 [57] and MS-COCO [58], which are often used for multi-label image classification tasks, were also selected as the datasets to carry out the comparison experiments. The above three datasets contain the amount of lost data ranging from small to large, and the generalization performance of the proposed method can be fully demonstrated by carrying out experiments on datasets of different sizes. It is worth noting that the VOC2012 dataset is richer in labeled categories than the other two datasets, which can effectively validate the model’s multi-label classification ability and is compatible with the subsequent more detailed label prediction of seismic damage categories.

Due to the lack of public datasets related to cultural relics for the assessment of the seismic damage state, in order to verify the practical effect of the proposed method, it is treated as the re-identification task to carry out comparative experiments according to the method principle. The re-recognition task assesses the similarity of the same target in two states, and its input and output data types are similar to those of the model. Therefore, the experiments use two classical re-identification datasets, VehicleID [59] and CUHK01 [60], in addition to the self-constructed CR-SDD dataset to evaluate the model. The CUHK01 dataset is similar in size to the CR-SDD dataset and can demonstrate the classification performance of the proposed method for the same size of data volume. While VehicleID is a high-quality, large-scale dataset, its experimental results can effectively prove the stability of the proposed method. According to the experimental requirements, since the degree of seismic damage of cultural relics is four categories, four different states are selected for each object in the dataset for model training. Among the several datasets selected in this study, the CUHK01 dataset is similar in size to the CR-SDD dataset, and datasets such as VOC2012 are larger. Despite the large differences in data scale size and classification objects, the validity and generalization of the method in this study can be verified side by side by carrying out comparative experiments on datasets of different sizes. All five kinds of data are used in the division ratio of 7:2:1 for model training, validation, and testing, as shown in Table 3.

The CR-SDD dataset contains data on artifacts in the collections of 56 museums in a province in Southern China. It contains basic attribute information, such as images; quality; and size of the front, side, and bottom surfaces of 1352 cultural relics, as well as information on the seismic damage of the relics through statistics. The dataset was labeled by several experts with the corresponding artifact image labels used for model training.

The VOC2012 dataset is a publicly available dataset used for image-processing tasks. It contains 11,530 images and 20 categories. Rich labeling information is provided in each image, including the location, size, and category of the target. In this study, this dataset is divided, and then its category information is used to conduct experiments.

The MS-COCO dataset is a large dataset built by Microsoft and used for image target detection and image-segmentation tasks. It contains 122,218 images consisting of 80 common categories with an average of 3 category labels per image. It has been widely used in multi-label image classification tasks.

The VehicleID dataset is a publicly available large-scale vehicle re-recognition dataset, which contains 26,267 vehicles, corresponding to a total of 221,763 images. According to the experimental requirements, only four sets of images for each vehicle are used for this experiment, totaling 106,058 photos of different angles of the vehicle.

The CUHK01 dataset is a publicly available pedestrian re-identification dataset. It contains 971 identities and 3884 images. The dataset is captured by two disjoint cameras with two images in each view, corresponding to four images for each identity.

### 5.2. Comparison Model and Evaluation Metrics

In order to verify the classification effect of the automatic matching model for cultural relics seismic data, four models, CPCL [61], SSGRL [62], MSRN [63], and DSDL [64], are used in this experiment to compare with A3MIF. The first three models all adopt similar attention mechanisms to accomplish the corresponding tasks by learning semantic tag word embeddings. CPCL models semantics in feature space through synthesis and decomposition operations, while SSGRL emphasizes exploring the semantic interactions between feature vectors through the attention mechanism. The MSRN approach makes full use of CNN to learn the semantic representation of images. DSDL implements multi-label image classification through dictionary learning, which introduces multi-source information fusion into the domain of image classification and serves as a comparison model because of its similarity to the task of the methods.

Since the reidentification task is used in the part of cultural relics seismic damage state assessment for comparison experiments, three models, SC-ReID [65], DDM [66], and MUSP [67], are chosen to compare with the ASMC proposed. Among them, SC-ReID is a method for personnel re-identification, which combines the attention mechanism to construct a two-branch deep feature learning architecture to realize personnel recognition. The latter two models are re-recognition methods used for vehicles, and both of them extract vehicle features by constructing deep learning architectures to realize target matching. It is worth noting that the outputs of the three re-recognition methods in this comparison experiment are no longer the similarity of the two images but the corresponding four categories (for the re-recognition dataset, different angles of the target object are regarded as one category).

The evaluation metrics on the VOC2012 dataset are the average precision (AP) of the 20 categories and the average precision (mAP) of all categories. It is calculated as shown in Equations (29) and (30).
(29)$${\mathrm{AP}}_{c}=\frac{1}{n}\sum_{i=1}^{n}\mathrm{precision}*(\mathrm{recall}(i))\;$$
(30)$$\mathrm{mAP}=\frac{1}{C}\sum_{c=1}^{C}{\mathrm{AP}}_{c}$$
where c represents the number of categories, n is the total number of predicted samples, $precision\left(i\right)$ represents the number of correctly predicted samples in the previous i samples divided by the total number of predicted samples, and $recall\left(i\right)$ represents the number of correctly predicted samples in the previous samples divided by the total number of actual samples.

We take the precision; recall; and F1 scores (P, R, and F1) for each category and the average precision, recall, and F1 scores (mP, mR, and mF1) for all categories as detailed performance metrics. For the CR-SDD dataset, since the labels to be categorized are the artifact type and damage degree, this experiment evaluates the labels with the top two prediction result scores for each image at the time of evaluation. For the MS-COCO dataset, on the other hand, multiple labels with prediction results of the top three are selected for evaluation. VehicleID and CUHK01 datasets are evaluated for the classification results of four categories. The formulas are shown in Equations (31)–(33).
(31)$$\mathrm{Precision}\;=\frac{\mathrm{TP}}{\mathrm{TP}+\mathrm{FP}}\;$$
(32)$$\mathrm{Recall}\;=\frac{\mathrm{TP}}{\mathrm{TP}+\mathrm{FN}}$$
(33)$$F1=\frac{2\times \;\mathrm{Precision}\;\times \;\mathrm{Recall}\;}{\;\mathrm{Precision}\;+\;\mathrm{Recall}\;}$$
where TP (true positives) denotes true positive category samples, and FP (false positives) denotes incorrectly predicting negative category samples as positive category samples. FN (false negatives) denotes incorrectly predicting positive category samples as negative category samples.

### 5.3. Evaluation Results of Seismic Damage State of Cultural Relics

Table 4 shows the evaluation results of the three models, SC-ReID, DDM, and MUSP, as well as the proposed ASMC on VehicleID and CUHK01 datasets. In the evaluation results, it can be seen that MUSP has the highest accuracy on the VehicleID dataset, which exceeds the proposed ASMC method by 2%. However, its evaluation results are lower than those of ASMC in both R and F1 values, which is analyzed because the MUSP model adopts the attention mechanism to focus on the local feature information, and it has a higher feature extraction and sensing ability. In the comparison experiments on the CUHK01 dataset, the P and F1 values are higher than the DDM model by 1.8% and 0.6%. Moreover, it performs best on the CR-SDD dataset. Meanwhile, we use the parameters and floating-point computations (FLOPs) to evaluate the complexity and execution efficiency of the model. In the results, it can be seen that the number of parameters of ASMC is slightly larger than that of DDM, but the FLOPs are the lowest. Combining the evaluation results of each index, the ASMC method proposed has strong classification performance and can realize the accurate assessment of the seismic damage status of cultural relics.

The trend of evaluation metrics of our proposed ASMC method with SC-ReID and other methods on three datasets, such as VehicleID, is shown in Figure 8, where red color represents accuracy, green color represents recall, and blue color represents the F1 value. The trend of the histogram shows that the two methods, MUSP and ASMC, perform better on the VehicleID dataset, but the histograms of the three metrics of ASMC are closer to each other, so its classification effect is more stable. In the comparison experiments on the CUHK01 dataset, the DDM model has higher values of P and F1, but the histograms representing the accuracy are lower, and there is a gap compared with ASMC. In contrast, ASMC performs the best on the CR-SDD dataset and has the closest indicator heights for the three colors. Combining the trends of the assessment indicators of each model, it can be concluded that our proposed method has a more stable classification performance and can accurately complete the assessment of the seismic damage status of cultural relics.

In order to verify the stability of the method, we compared the loss curves of four models, such as ASMC, on the CR-SDD dataset, as shown in Figure 9. The experimental results demonstrate that the ASMC method has a faster convergence rate, especially in the first 10 cycles, where the loss decreases rapidly and then gradually stabilizes. The other methods also converge equally fast, but their curves have large fluctuations and slower convergence speeds, proving that ASMC is more likely to achieve the best relics class classification.

### 5.4. Evaluation of the Effect of Matching Data on Cultural Relics’ Seismic Damage

Although the current cultural relics seismic damage label types are relatively small, with the breakdown of cultural relics types and seismic damage levels, the label types have been increased and improved. Therefore, in order to assess the comprehensive effect of the A3MIF method in the label prediction of cultural relics seismic damage data, we analyzed both the small number of label prediction and multi-label classification tasks. Among them, the experiments for both CR-SDD and MS-COCO datasets have a small number of label predictions, while the evaluation using the VOC2012 dataset is a multi-label classification task.

#### 5.4.1. Evaluation of Label Matching for Relic Seismic Data

The label prediction results of the A3MIF model on the CR-SDD dataset are shown in Table 5. A3MIF obtained the highest mF1 value, up to 82.6, among all the compared methods and the best evaluation scores on mF1 and mR. Performance gains of 7.1%, 1.7, and 7.4% were achieved compared to CPCL, SSGRL, and MSRN, respectively. The reason for this is analyzed because the semantically guided attention mechanism improves the learning ability of the model, and the obtained image features contain relatively less interfering information. Similarly, a performance gain of 3.6% is achieved on mP compared to DSDL using dictionary learning. The performance gain, on the other hand, is due to the fact that the embedded learning of semantic dictionaries plays a more important role in the face of the artifact seismic damage dataset A3MIF, which is able to capture the common relationships between complex labels.

Figure 10 shows the trend of the label prediction evaluation results of our proposed ASMC method with CPCL and other methods on the CR-SDD dataset, where blue color represents the average accuracy of the two types of labels, orange color represents the average recall, and green color represents the average F1 value. As can be seen in the trend of the histograms, the histograms of the three indicators of the A3MIF method on the CR-SDD dataset are the highest, and therefore it has the best label prediction results. It can accurately complete the label matching of cultural relics’ seismic data.

The label prediction results on the MS-COCO dataset for each model are shown in Table 6. Facing a much larger and more complex dataset, A3MIF achieves the highest mP among all the compared methods, with 8.3%, 3.5, and 9.3% gains compared to CPCL, SSGRL, and MSRN, respectively. A performance gain of 2% in mP was achieved compared to the DSDL method. The reason for analyzing the more significant performance gain of the A3MIF model on larger datasets is that semantic features also accompany larger training data. Meanwhile, the attention mechanism in A3MIF enables better-embedded learning of complex semantic dictionary labels. In addition, the massive image features contain relatively independent semantic information, which helps to better optimize the model parameters when classifying label predictions.

Figure 11 shows the trend of the three labeled prediction assessment results of our proposed ASMC method with CPCL and other methods on the MS-COCO dataset. As can be seen in the trend of the histogram, the three indicators of the A3MIF method on the MS-COCO dataset are the highest compared to the other methods, and the height of the indicators of the three colors is the closest to each other compared to the other models. Therefore, its prediction effect is also more stable, and it also has strong prediction performance in the three-label matching.

In order to comprehensively evaluate the prediction of the above five methods on two small numbers of labeled datasets, we averaged the mP, mR, and mF1 for each method and plotted the trend of change, as shown in Figure 12. Based on the variation of the histograms, it can be seen that the combined performance of the methods proposed is the best on both datasets. The average mP value and mF1 are 2.6% and 6.35% higher than the best SSGRL method, and the average mR value is 5.42% higher than the MSRN method. The experimental results demonstrate that the A3MIF method has high accuracy in label prediction for a small number of samples oriented to multi-source data.

#### 5.4.2. Analysis of Multi-Label Classification Prediction Results

Our proposed A3MIF model was compared with four other multi-label image classification methods on the VOC2012 dataset, and the average classification accuracies of the methods on 20 categories (aero, bike, bird and boat, etc.) are shown in Table 7. A3MIF performs much better on this dataset, with a 0.2% improvement in mAP over the best-performing DSDL method, which uses dictionary learning, and a 2.71% improvement over the other methods. The DSDL method, which is the best-performing method and uses dictionary learning, improves by 0.2% on mAP and 2.71% on average over the other methods. A3MIF achieves the first AP score on 16 categories, and its AP score is lower than that of other methods by a small margin on only 4 categories. The experimental results fully prove that the method proposed can fully integrate the semantic information of multi-source data, has a better recognition effect when dealing with multi-label classification tasks, and can provide model support for the multi-label classification task of cultural relics’ seismic damage dataset.

Figure 13 shows the multi-label prediction effect of the five methods on the VOC2012 dataset. According to the change of the fold line, it can be seen that our proposed method has the best accuracy in most of the category’s predictions, but it is slightly lower than the DSDL method in the label prediction of some categories. The reason for this is that A3MIF focuses more on the embedding and learning of semantic information, which obliterates some of the detailed features of the image when fusing them with the visual semantic information embedded in the image, resulting in biased prediction results. Considering that the seismic damage label information is more complex and the semantic priority is higher than that of images, the label-prediction performance of the A3MIF method should be more significant when matching the seismic damage dataset for cultural relics.

### 5.5. Calibration of Seismic Damage Dataset of Cultural Relics

According to the collected cultural relics collection space, ontological attributes, and seismic damage information, we adopted the MySQL database to store the relevant numbers and used the Java language to create a museum website for visualization. Figure 14 shows the effect diagram of the visualization webpage of the seismic damage information of cultural relics. According to the function panel in the figure, the name of the cultural relics can correspond to the query of the basic attributes of the cultural relics and seismic damage information. The newly collected data on cultural relics can be supplemented by the add button. In addition, wrong data items can be modified.

Based on the analysis of the above experimental results, it can be seen that we use manual measurements and other ways to count the basic attributes of cultural relics and design a multi-source feature-fusion matching method based on deep learning to realize the accurate assessment of cultural relics’ damage status and automated matching of labels. Compared with multiple comparison models, our proposed method performs the best and can effectively ensure the accuracy of the seismic damage data of cultural relics.

Since the seismic damage dataset of cultural relics we collected and constructed contains a large amount of textual and numerical information, in order to ensure the reliability and accuracy of the dataset, it is necessary to validate the collected and calculated cultural relics data. Since the volume difference of movable cultural relics is small and the related attribute values are relatively close, we adopted the box-plot method to validate the numerical data. We selected the cultural relics body and display facilities attribute data items for verification, mainly including the length, width, height, caliber, mass, static friction coefficient (u0), and power amplification factor.

If there are outliers in the data items, they are usually represented as separate points in the graph outside the upper and lower boundaries of the box. If there are no outliers, the box plot will be presented as a neat box shape. The upper and lower boundaries of the box are determined by the upper quartile and lower quartile of the data, respectively, and the median is located in the center of the box. There will be no individual points located outside the boundaries of the box. As can be seen in the results shown in the box plot drawn from the seismic data of the cultural relics in Figure 15, the seismic data of the cultural relics obtained from the acquisition have no outliers, and they are all within the normal range.

## 6. Conclusions

Aiming at the problem of scarcity of real data and imperfect processing strategy of seismic damage of cultural relics in collections, we propose a multi-source feature-fusion method based on deep learning to determine the seismic damage data of cultural relics in collections. This study not only utilizes a large number of sensor devices but also involves the collection, processing, and analysis of image data. Firstly, the multi-source data processing strategy of seismic-damaged cultural relics is formulated to achieve standardized information acquisition and processing. We define the event-ontology model that contains the associated information of seismic-damaged cultural relics and design a standardized empirical model to solve the information about the dynamic coefficient of the relics with the center of mass and other information about the preservation space and seismic. We use it as a universal and standard statistical index system for seismic-damaged data of the cultural relics in the collection. Secondly, a deep learning-based multi-source feature-fusion matching method is proposed to realize efficient data annotation. Based on the images of cultural relics before and after the seismic event, we constructed a damage state assessment model of cultural relics by fusing superpixel map convolution to classify the damage level of cultural relics. Combined with deep learning, we proposed a multi-source data matching method for seismic-damaged cultural relics to realize the automated fusion and matching of seismic-damaged cultural relics data. Finally, for the existing 56 cultural relics custodian units, we used the above multi-source data-processing strategy and automated information-fusion method to integrate the seismic damage information of 1352 cultural relics in the collection.

It is shown by comparing various classification models on different datasets that the accuracy of the multi-source feature fusion of cultural relic earthquake data based on deep learning is up to 93.6. The accuracy of cultural relics label matching is as high as 82.6%. It is clear that our proposed method also achieves better results by analyzing the stability and complexity of various models. Meanwhile, we construct a complete dataset for seismic damage risk analysis of cultural relics in the collection, which will provide a data basis for the subsequent research of seismic damage impact analysis of cultural relics in the collection. Our proposed method effectively improves the science and effectiveness of cultural relics protection through the Internet of Things and artificial intelligence technology. But what we have used is only a small part of it. Therefore, considering the use of more sensors to monitor the status of cultural relics in the next step of the research will greatly improve the automation and intelligence level of cultural relic protection, which in turn will provide a strong guarantee for the long-term preservation of cultural relics.

## Figures and Tables

**Figure 1 sensors-24-04525-f001:**
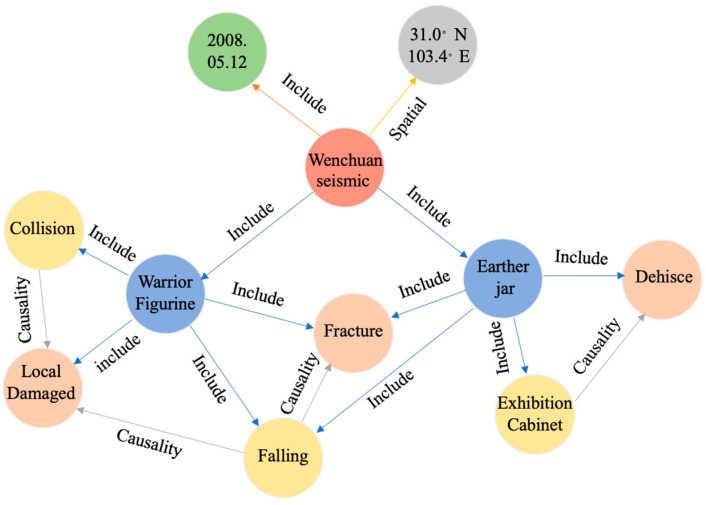
Event-ontology model for seismic impact analysis of curatorial relics (EOM-SR).

**Figure 2 sensors-24-04525-f002:**
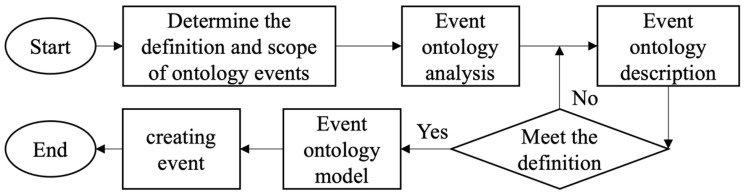
EOM-SR creation flowchart.

**Figure 3 sensors-24-04525-f003:**
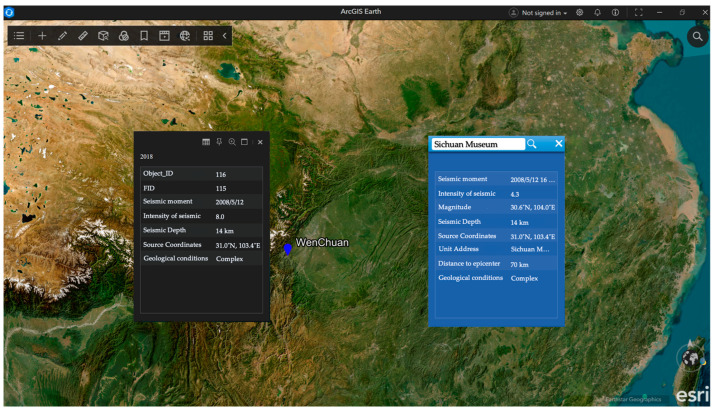
AriGIS seismic data visualization display.

**Figure 4 sensors-24-04525-f004:**
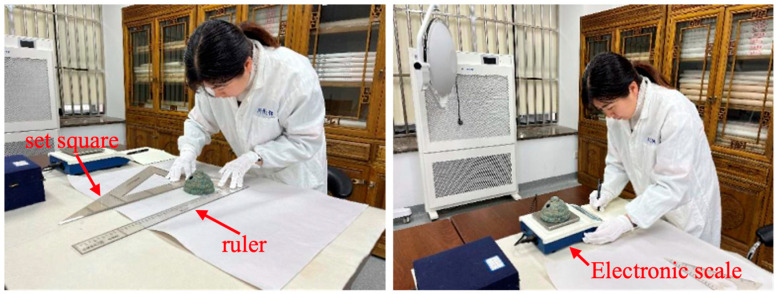
Cultural relics ontology data-acquisition process diagram (the left side shows the artifact size measurement method; the right side shows the artifact quality measurement method).

**Figure 5 sensors-24-04525-f005:**
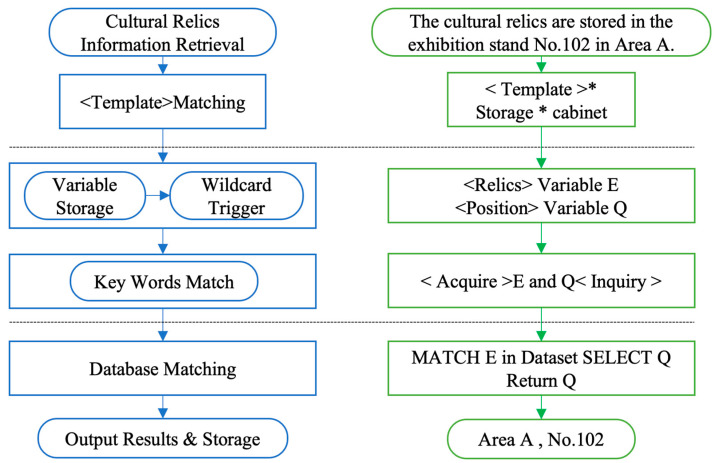
Architecture diagram of template matching query for cultural relics exhibition and collection facility information.

**Figure 6 sensors-24-04525-f006:**
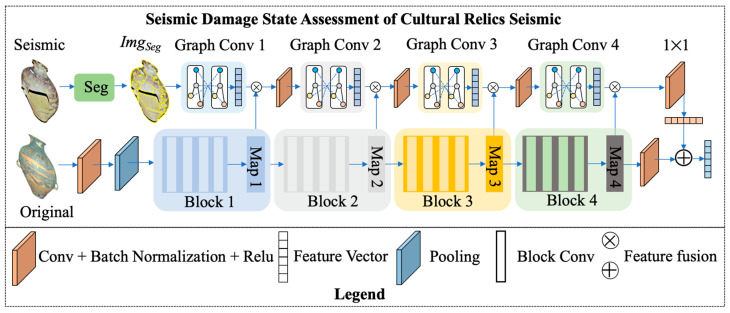
Architecture diagram of seismic damage state assessment of cultural relics by fusing superpixel map convolution.

**Figure 7 sensors-24-04525-f007:**
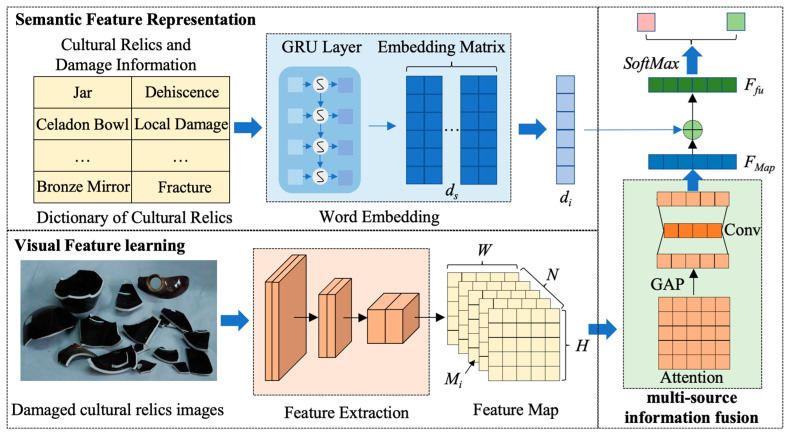
Architecture diagram of automatic matching of cultural relics seismic damage data based on multi-source information fusion. The dictionary of cultural relics seismic damage data is shown in Appendix A, Table A1.

**Figure 8 sensors-24-04525-f008:**
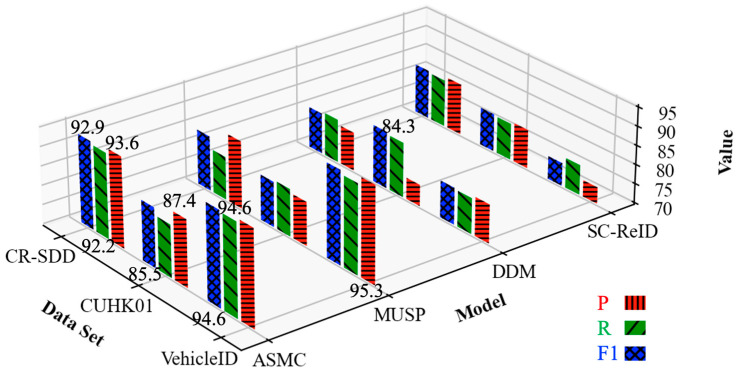
Comparison of trends in indicators used for assessing the seismic condition of relics.

**Figure 9 sensors-24-04525-f009:**
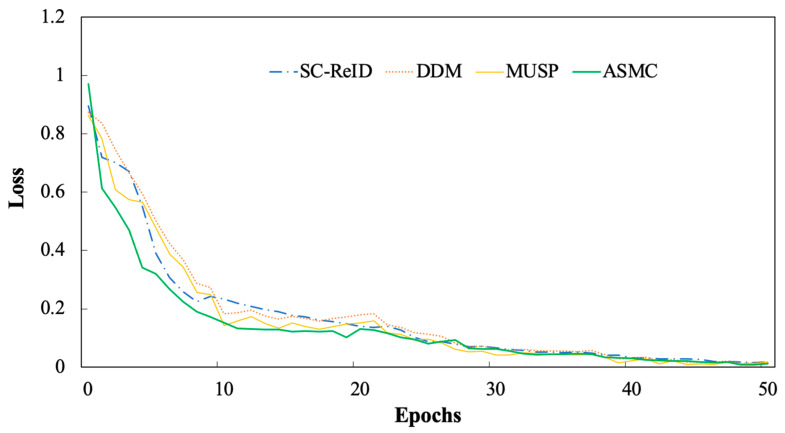
Model training loss variation curve.

**Figure 10 sensors-24-04525-f010:**
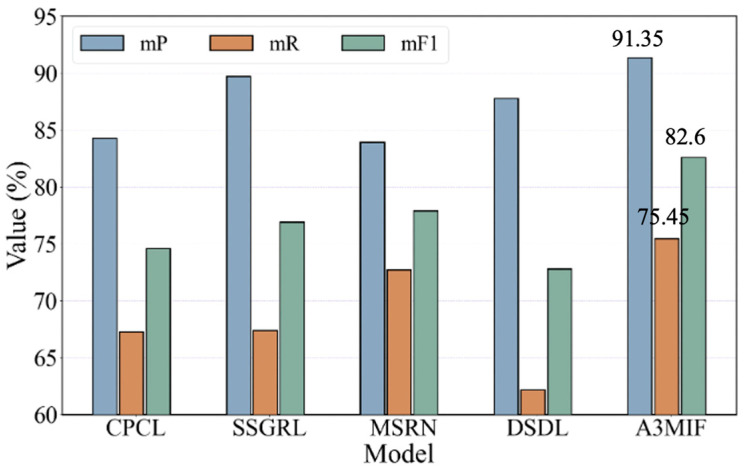
Comparison of trends in classification prediction results of CR-SDD.

**Figure 11 sensors-24-04525-f011:**
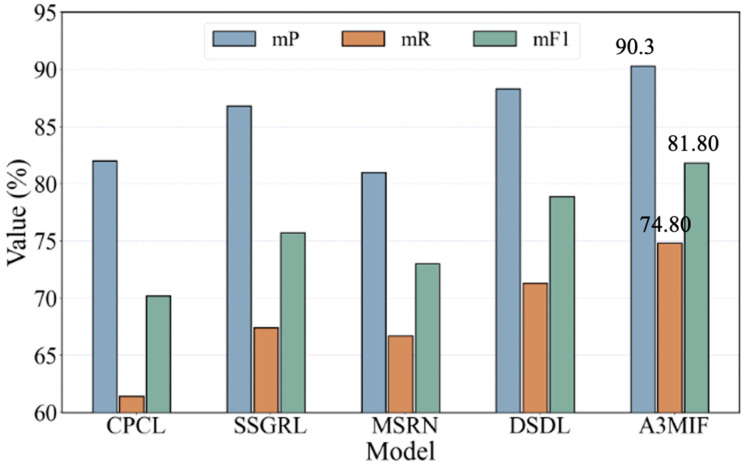
Comparison of trends of classification prediction results using MS-COCO.

**Figure 12 sensors-24-04525-f012:**
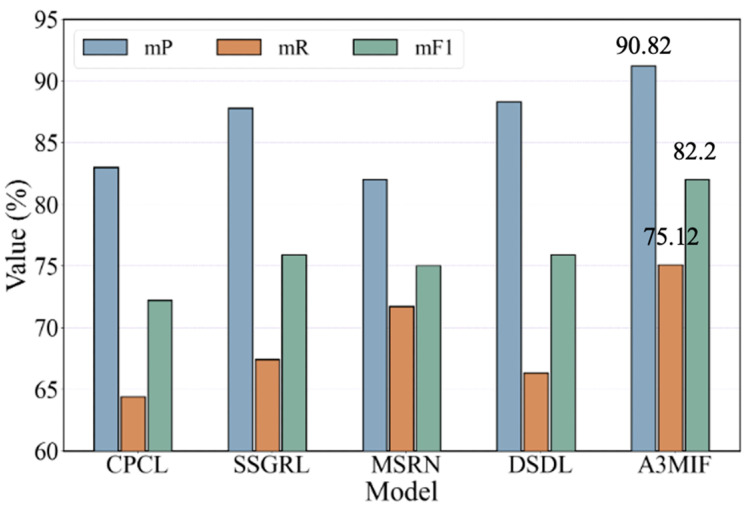
Average results of multi-model label prediction using both datasets.

**Figure 13 sensors-24-04525-f013:**
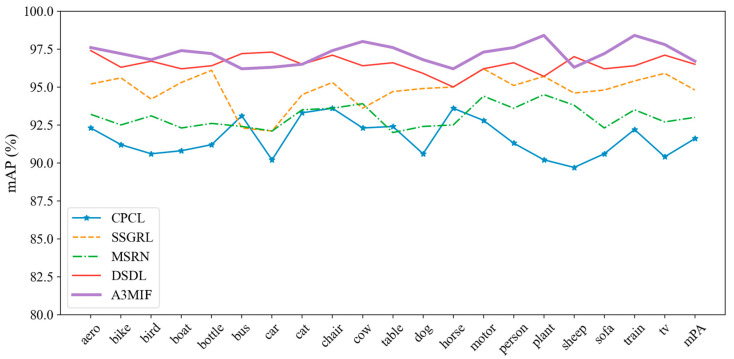
Multi-model labeling prediction results of VOC2012.

**Figure 14 sensors-24-04525-f014:**
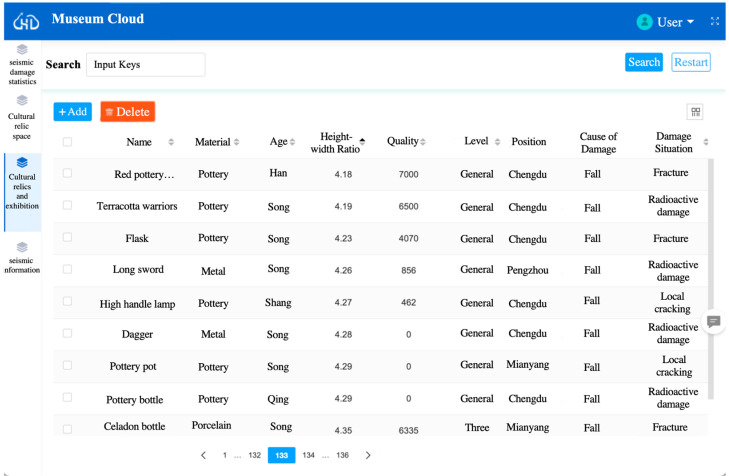
Visualization webpage of seismic damage dataset of cultural relics.

**Figure 15 sensors-24-04525-f015:**
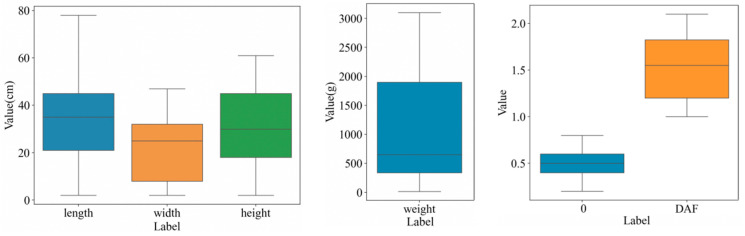
Attributes of the cultural relic ontology and exhibition facilities’ box-type diagrams.

**Table 1 sensors-24-04525-t001:** Definition of event attributes for seismic impact analysis of cultural relics in collections.

No.	Attributes	Definition	Description
1	subject	Event Initiator	Seismic Disasters
2	Object	The recipient of the event	Seismic Artifacts
3	Influence	Correlating factors contributing to the occurrence of the event	Seismic Impact Factor
4	Time	The point in time when the event occurs	Seismic events
5	Space	The physical space in which the event occurred	Location of the seismic
6	Event Type	Types of damage to artifacts during the event	Types of Damage to Cultural Objects
7	Causality	The role of the events in relation to each other	Relationships that cause damage to artifacts
8	Include	The attribution of the events, i.e., Event B is a sub-event of Event A.	Inclusion of seismic influences
9	Temporal	The sequence of events	The process of seismic damage to cultural objects
10	Spatial	The spatial correlation between the events	Physical relations in the space where the seismic damage to cultural objects occurred

**Table 2 sensors-24-04525-t002:** Experimental environment and parameter configuration.

Environment	Configuration	Model Parameters	Configuration
GPU	RTX3080 (10 GB) * 1	weights_init	xavier
OS	ubuntu20.04	optimizer	Adam
Framework	PyTorch1.11.0	Batch_size	16
Language	Python3.8	lr	1 × 10^−3^
cuda	11.3	epoch	50

**Table 3 sensors-24-04525-t003:** Experimental dataset configuration.

DataSet	Train	Validation	Test
CR-SDD	946	270	136
VOC2012	8071	2306	1153
MS-COCO	85,552	24,443	12,223
VehicleID	74,240	21,212	10,606
CUHK01	2720	7,76	388

**Table 4 sensors-24-04525-t004:** Evaluation results of seismic state of cultural relics using ASMC and other methods.

Method	VehicleID			CUHK01			CR-SDD			Params (M)	FLOPs (G)
	P	R	F1	P	R	F1	P	R	F1		
SC-ReID	74.3	77.8	76.0	80.2	79.6	79.9	83.5	82.4	82.9	25.6	4.1
DDM	80.1	79.3	79.6	85.6	84.3	84.9	80.1	81.3	80.7	19.2	2.4
MUSP	95.3	93.3	94.3	81.4	82.3	81.8	87.6	81.2	84.3	26.6	3.5
ASMC	95.1	94.2	94.6	87.4	83.6	85.5	93.6	92.2	92.9	20.6	2.3

**Table 5 sensors-24-04525-t005:** Prediction results for the two types of labels on CR-SDD.

Method	Classes	Evaluation Metrics (Per)			Evaluation Metrics (Average)		
		P	R	F1	mP	mR	mF1
CPCL	1	87.1	63.3	73.3	84.25	67.25	74.6
	2	81.4	71.2	75.9			
SSGRL	1	90.1	63.2	74.3	89.7	67.4	76.9
	2	89.3	71.6	79.4			
MSRN	1	83.2	74.1	78.3	83.9	72.7	77.9
	2	84.6	71.3	77.3			
DSDL	1	88.0	63.2	73.5	87.75	62.15	72.8
	2	87.5	61.1	72			
A3MIF	1	90.6	76.3	82.8	91.35	75.45	82.6
	2	92.1	74.6	82.4			

**Table 6 sensors-24-04525-t006:** Prediction results for the three classes of labels using MS-COCO.

Method	Classes	Evaluation Metrics (Per)			Evaluation Metrics (Average)		
		P	R	F1	mP	mR	mF1
CPCL	1	82.0	60.6	69.7	82.0	61.4	70.2
	2	83.5	62.2	71.3			
	3	80.3	61.4	69.6			
SSGRL	1	89.6	64.3	74.9	86.8	67.4	75.7
	2	88.7	67.6	76.7			
	3	82.1	70.3	75.7			
MSRN	1	80.1	65.5	72.1	81.0	66.7	73.0
	2	79.3	70.3	74.5			
	3	83.5	64.4	72.7			
DSDL	1	88.3	70.2	78.2	88.3	71.3	78.9
	2	89.1	72.2	79.8			
	3	87.4	71.4	78.6			
A3MIF	1	91.1	76.2	82.9	90.3	74.8	81.8
	2	90.3	78.1	83.8			
	3	89.6	70.1	78.7			

**Table 7 sensors-24-04525-t007:** Prediction results for 20 classes of labels on the VOC2012.

Classes	Method				
	CPCL	SSGRL	MSRN	DSDL	A3MIF
aero	92.3	95.2	93.2	97.4	97.6
bike	91.2	95.6	92.5	96.3	97.2
bird	90.6	94.2	93.1	96.7	96.8
boat	90.8	95.3	92.3	96.2	97.4
bottle	91.2	96.1	92.6	96.4	97.2
bus	93.1	92.3	92.4	97.2	96.2
car	90.2	92.1	92.1	97.3	96.3
cat	93.3	94.5	93.5	96.5	96.5
chair	93.6	95.3	93.6	97.1	97.4
cow	92.3	93.6	93.9	96.4	98.0
table	92.4	94.7	92.0	96.6	97.6
dog	90.6	94.9	92.4	95.9	96.8
horse	93.6	95.0	92.5	95.0	96.2
motor	92.8	96.2	94.4	96.2	97.3
person	91.3	95.1	93.6	96.6	97.6
plant	90.2	95.7	94.5	95.7	98.4
sheep	89.7	94.6	93.8	97.0	96.3
sofa	90.6	94.8	92.3	96.2	97.2
train	92.2	95.4	93.5	96.4	98.4
tv	90.4	95.9	92.7	97.1	97.8
mPA	91.6	94.8	93.0	96.5	97.2

## Data Availability

The datasets presented in this article are not readily available because the data are part of an ongoing study or cannot be provided due to technical reasons. Requests to access the datasets should be directed to https://www.kaggle.com/datasets/joybrosion/cultural-relic-dataset-part (accessed on 2 July 2024).

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
