# Peer review of "Multi-Source Feature-Fusion Method for the Seismic Data of Cultural Relics Based on Deep Learning"

_sensors, 2024, doi:10.3390/s24144525_

Round 1
Reviewer 1 Report
Comments and Suggestions for Authors
In the study, multi-source data processing strategy is developed according to the needs of seismic impact analysis of the cultural relics in the collection, and the seismic event ontology model of cultural relics is constructed,which is commendable. However, to further enhance the manuscript's clarity and contribute to the field's advancement, the following points should be addressed:
1. Section 2.1 appears twice. Please revise to ensure unique section numbering and re-evaluate the manuscript for any other inconsistencies.
2. Line 258 references "56 units," but the source of this data is not provided. Please include a citation or explanation for it.
3. The data format normalization method or standard mentioned in line 334 is not specified. Providing explicit details on the normalization process will improve the manuscript's clarity.
4. In Section 4.1, the effectiveness of the cultural relic damage state assessment and examples are mentioned but not reflected in subsequent sections. Additionally, the damage state levels mentioned in lines 427-428 are not explained in the text. Including this information is crucial for reader comprehension.
5. An explanation of the loss function used by the neural network is recommended in Section 4.2. Detailing the loss function will provide better insight into the model's training process.
6. Details about the Seismic Damage Dataset of Cultural Relics (CR-SDD) referenced in line 493 should be provided. This information is necessary to understand the dataset's composition and its relevance to the study.
7. In Section 5.3, a comparison of ASMC with other neural networks in terms of parameter count (Params) and computational complexity (FLOPs) is recommended. Discussing whether the accuracy improvements result in excessive computational costs or prolonged training and inference times will provide a balanced evaluation of the model's performance.
8. Chapter 5 should highlight the cultural relic dataset (CR-SDD) and the damage assessment method (ASMC) used in the damage assessment process. Analyzing the model using loss function curves and other evaluation metrics (Precision, Recall, etc.) will enhance the manuscript's analytical depth.
9. The manuscript mentions the use of various measurement sensors in lines 303 and 742, but does not specify the types of sensors or their usage.
10. The proposed method of multi-source feature fusion of cultural relic earthquake data based on deep learning needs to be clarified. Highlighting its advantages and disadvantages compared to similar methods will underscore the novelty and relevance of the proposed approach.

The English language in the manuscript is clear and understandable. However, it could benefit from a more concise and direct style. Additionally, attention to grammar and punctuation would enhance readability. Overall, improvements in language precision and structure could elevate the quality of the manuscript.
Author Response
Dear editor:
Thank you very much for your suggestions, we have revised the article according to your comments. Attached is a specific response to your comments, please attention to check.
Best wishes!

Reviewer 2 Report
Comments and Suggestions for Authors
The manuscript "Multi-source Feature Fusion Method for Seismic Data of Cultural Relics based on Deep Learning" presents a comprehensive approach to addressing the challenges associated with seismic hazard prevention and protection of cultural relics. The authors have developed a deep learning-based method to improve the analysis and processing of seismic damage data for cultural relics in collections. Here are some suggestions for the authors:
- While the authors have created a dataset, expanding this with more examples, especially from different geographical locations and cultural contexts, could improve the model's generalizability.
- It might be beneficial to discuss the potential for applying this method to long-term monitoring of cultural relics, possibly through the integration of IoT devices and continuous data collection.
- The article could benefit from a section discussing how cultural heritage professionals and conservators might engage with the technology and dataset.
- Given the use of cultural relics data, it would be important to address any ethical considerations, particularly around data privacy and the respectful representation of cultural heritage.
- While the method shows high accuracy, insights into its scalability and computational efficiency could be valuable, particularly for smaller institutions with limited resources.
- Given the high stakes in cultural heritage protection, understanding how the model performs in terms of false positives and the measures taken to validate its predictions would be important.
- Discussing how this method could be integrated with existing museum management and disaster preparedness systems could be beneficial.
Moderate editing of English language required.
Author Response
Dear expert:
Thank you very much for your suggestions, we have revised the article according to your comments. Attached is a specific response to your comments, please attention to check.
Best wishes!

Reviewer 3 Report
Comments and Suggestions for Authors
The paper is well-organized and provides sufficient details on the proposed multi-source feature fusion method for seismic data of cultural relics based on deep learning. However, there are a few areas that could benefit from further elaboration.
- Real-Time Analysis and Rapid Response: The authors mention the need for real-time analysis and rapid response but do not provide any details regarding the computing time or the speed of the proposed method. It would be beneficial to include a comparison of the method's computational time against other existing methods. Specifically, what is the fusion speed of your method? How quickly can it process and analyze seismic damage data?
- Dimension Reduction Methods: What is the size of features? Clarify if any dimension reduction methods were employed in the proposed approach. If not, consider discussing why these methods were not used. If dimension reduction methods were used, provide details on the methods (e.g., eigenvectors) and compare their results with the proposed method in terms of both accuracy and efficiency.
- Comparative Analysis: While the accuracy of the method is highlighted, a comparative analysis involving both accuracy and time cost should be presented. How does the proposed method compare with existing earthquake damage state assessment models in terms of processing speed and resource efficiency? This would provide a more comprehensive evaluation of the method's performance.
Author Response

(The authors gave the same response as above.)

Round 2
Reviewer 1 Report
Comments and Suggestions for Authors
The manuscript has been revised according to the reviewer's comments. The reviewer does not have any further comments.